# A Pangenomic Approach to Improve Population Genetics Analysis and Reference Bias in Underrepresented Middle Eastern and Horn of Africa Populations

**DOI:** 10.3390/biom15040582

**Published:** 2025-04-15

**Authors:** Adrien Oliva, Rachel Foare, Peter Campbell, Natalie A. Twine, Denis C. Bauer, Angad Singh Johar

**Affiliations:** 1Australian e-Health Research Centre, Commonwealth Scientific and Industrial Research Organisation (CSIRO), Melbourne 3169, Australia; natalie.twine@csiro.au (N.A.T.); denis.bauer@csiro.au (D.C.B.); 2Life Sciences and Health Graduate School, Université Paris-Saclay, 3 Rue Joliot Curie, 91190 Gif-sur-Yvette, France; rachel.foare@gmail.com; 3Information Management and Technology (IM&T), Commonwealth Scientific and Industrial Research Organisation (CSIRO), Melbourne 3169, Australia; peter.h.campbell@csiro.au; 4Menzies Institute of Medical Research, The University of Tasmania, Hobart 7000, Australia; angad.johar@utas.edu.au

**Keywords:** pangenomics, variation graph, reference bias, genomic medicine, population genetics, underrepresented populations

## Abstract

Genomics plays a crucial role in addressing health disparities, yet most studies rely on the hg38 linear reference genome, limiting the potential of pangenomic approaches, particularly for underrepresented populations. In this study, we focus on characterising East African populations, particularly Somalis, by constructing a variation graph using Mozabites from the Human Genome Diversity Project (HGDP) given their ancestral affinity with Somalis. We evaluated the effectiveness of this graph-based reference in estimating effective population sizes (*Ne*) in Bedouins compared to the hg38 reference and examined its impact on allele frequencies and genome-wide association studies (GWAS). Applying a coalescent model to the graph-based reference produced a *Ne* estimate of approximately 17 for the Bedouin population, which was significantly lower than the estimate from the hg38 reference (approximately 79,000). Only the graph-based estimate fell within the 95% confidence interval in simulations, indicating improved accuracy. Moreover, graph variants exhibited significantly lower allele frequencies (*p*-value < 2.2 × 10^−16^), suggesting potential effects on the interpretation and power of GWAS. Notably, GWAS variants specific to Bedouins derived from the graph showed lower frequencies (*p* = 0.023) than those obtained from the linear reference. These findings suggest that a pangenomic approach, informed by populations with ancestral affinities such as the Mozabites, provides more accurate estimates of *Ne* and allele frequencies. This highlights the importance of pangenomic strategies to better capture genetic diversity in underrepresented populations, a critical step towards improving population genetics studies, personalised medicine, and equitable healthcare.

## 1. Introduction

Northern and Eastern Africa, along with the Near East, have played a pivotal role in human evolution, shaping both genetic diversity and phenotypic variation. These regions served as a crucial corridor for the initial Out-of-Africa (OoA) migrations, significantly influencing the genetic makeup of modern humans [1]. However, the precise mechanisms through which such demographic processes have impacted present-day genetic variation remain poorly understood. The demographic history of these regions is highly complex and not yet fully deciphered, posing challenges for identifying the genomic variation underlying complex phenotypes. Recent studies provide strong evidence that these populations were gateways for OoA migrations, as demonstrated by haplotype similarities and by population split times estimated using Multiple Sequentially Markovian Coalescent (MSMC) models [2]. However, since the Neolithic period, these regions have also undergone extensive back migrations over the last 38,000 years [3] from various Levantine and Anatolian-based source populations [3,4], predating later waves of migration associated with the Roman Empire and Rashidun Caliphate.

The Horn of Africa (HOA), including Somalia and Ethiopia, has been at the intersection of both within-continent and transcontinental migrations. Within Africa, it has experienced gene flow from Sub-Saharan populations and the Maghreb, while external influences came from the Near East and Middle East. Notably, in Neolithic times (~6000–7000 years ago), populations from modern-day Israel and Palestine contributed to the HOA gene pool, followed by later migrations from the Rashidun Caliphate (~7th century CE) originating in present-day Saudi Arabia [2,3]. These migration events were interspersed with periods of isolation and genetic drift [2,3]. Understanding this complex genetic history requires a well-integrated approach that incorporates genomic data and archaeological research.

Early studies have provided insights into population dynamics after the initial OoA migrations, particularly the bottlenecks that occurred as humans settled in Arabia [5] for approximately 30,000 years prior to the accelerated dispersal across Eurasia [6]. According to the “Arabian Standstill Hypothesis” [7], the Bedouins are direct descendants of these early settlers [6]. They represent the earliest population carrying the Basal Eurasian lineage after the primary OoA exits and have experienced prolonged periods of isolation that contributed to their high levels of genetic homogeneity and North African ancestral components (17%) [8]. Following their migration to Kuwait and Israel’s Negev Desert, the Bedouin maintained much of their original genomic structure, largely due to their sociopolitical organisation and cultural practices [8]. However, the genetic composition distinguishing the HOA populations in our study was shaped by later historical events, particularly from the late Paleolithic and Neolithic periods onwards [9].

By the Neolithic Period, human populations had expanded beyond the Sinai Peninsula into the broader Near East and Middle East. One of the most prominent groups to flourish in this region was the Levantine Neolithic farmers, whose ancient DNA is primarily associated with populations from present-day Israel. Their genetic makeup reflects a strong continuity with local Natufian hunter-gatherer predecessors [9,10]. Although some studies suggest shared genetic lineages between Levantine Neolithic farmers and Bedouin groups [11], these populations remain genetically distinct. Additionally, Levantine Neolithic farmers are considerably contrasted in their evolutionary dynamics from other contemporary cultures in the Near East and the rest of Eurasia in terms of their drift and admixture patterns. One obvious distinction is their level of Basal Eurasian genomic content and degree of Neanderthal introgression, as well as their mixing with other Eurasian hunter-gatherers compared to other contemporary groups in the Near East and Europe [9]. The unique cultural and genetic identities of these early populations are further supported by archaeological evidence, including distinct pottery designs, burial practices, and other material artifacts, which reinforce their differentiation from one another.

In addition to shaping the demographic history of later Near Eastern populations, Levantine Neolithic groups also contributed genetically to both North African populations (e.g., ancestors of the Mozabites in Algeria) and Eastern Horn of Africa (HOA) ethnolinguistic groups, primarily Somalis and Ethiopians [3,12]. Genetic and archaeological evidence suggests that early back migrations from the Near East into Africa occurred approximately 23,000–30,000 years ago (kya), based on F_ST_ divergence estimates and material culture data. However, subsequent gene flow from the Levant around 6000–7000 kya also played a significant role in shaping the genetic makeup of Mozabites in the Maghreb and HOA populations [3,12]. Over this extended period (~30 kya, predating the Bantu expansion ~3 kya), Levantine Neolithic individuals likely intermixed with Epipaleolithic North African ancestors of the Mozabites and with early East African farming communities [13,14]. Both of these African groups, Tamazight-speaking North Africans and Cushitic-speaking HOA populations, belong to the Afro-Asiatic language family. The Cushitic speakers initially expanded as far south as Kenya and northern Tanzania, prior to the Bantu expansion (~1000 BCE) [15,16]. Following this large-scale migration, their geographic distribution became primarily restricted to Somalia and Ethiopia [17].

As successive migrations introduced greater genetic heterogeneity, particularly with the influx of Basal Eurasian descendants, genetic drift continued to shape the population structure of these African regions. This process occurred alongside multiple admixture events, further refining the distinct genetic profiles observed today. One key distinction is the higher genetic differentiation within and between HOA populations compared to Mozabites and other Maghrebi groups [12]. Recent studies also show that, despite geographic proximity, different North African ethnic groups exhibit substantial variation in Identity by Descent (IBD) and effective population size (*Ne*), highlighting complex demographic histories [18,19]. While gene flow between North Africa and the HOA did occur, its patterns varied significantly across ethnolinguistic groups. As a result, a unique indigenous-non-indigenous ancestry referred to as the Ethio-Somali component emerged within HOA populations. This component is highly differentiated from the rest of Africa, shaped by both the timing and sources of admixture [12]. The distinct genomic architecture of HOA populations, especially Somalis, has been further reinforced by long-term endogamy and cultural isolation [20]. Within Somali populations, later migrations from the Arabian Peninsula, long after the ancient Back-to-Africa migrations, contributed to cultural shifts and modulated population divergence [12]. However, despite these more recent influences, pre-agricultural evolutionary processes appear to have played a far more significant role in shaping the genetic landscape of both Somalis and the broader HOA populations, persisting from ancient times to the present [12].

Understanding demographic history is essential for improving genetic epidemiology and personalised medicine, particularly in understudied populations such as Somalis, Mozabites, and Bedouins. These groups experience some of the highest global mortality rates from cardiovascular disease [21,22], yet remain underrepresented in genomic research. A major limitation in genetic risk prediction is the poor transferability of polygenic risk scores (PRS) from European to non-European populations. Even within Europeans, population substructure significantly affects PRS accuracy [23,24,25]. The most notable example comes from studies by Berg et al. 2019 and Sohail et al. 2019 [23,24], which found that many genome-wide association study (GWAS) signals were spurious. These errors were caused by strong correlations between PCA loading scores, allele frequency differences, and effect sizes across Northern and Southern Europeans [23,24,25]. Such biases highlight how phenotypic gradients and allele frequency differences can create false-positive associations, undermining PRS reliability.

Before considering demographic history, though, another critical issue in human evolution and genetic epidemiology is reference bias. This occurs when genetic variation from non-European populations is poorly represented, leading to an overrepresentation of the reference allele compared to the alternative allele for any reference such as hg38 and CHM13v2. The root of this problem lies in the composition of reference genomes, which are over 70% derived from individuals of European ancestry. As a result, studies have reported that up to 10% of sequence reads from African populations, primarily West and Central Africans, fail to map correctly to these references [26]. This issue is likely even more pronounced for populations whose genetic variation remains poorly characterised, such as Eastern and Northern Africans [27]. Consequently, their sequence reads are at greater risk of misalignment, reducing the accuracy of genetic analyses compared to European populations.

Addressing this critical gap requires accurate and unbiased genomic data to identify genetic variants linked to complex traits, such as cardiovascular disease, while also gaining deeper insights into the fine-scale demographic history of underrepresented populations. In this study, we compare the performance of a pangenome-based variation graph [28] against the traditional linear reference genome (hg38) in its ability to infer genetic history and disease etiology. Specifically, we focus on a Middle Eastern population (Bedouin from the Human Genome Diversity Project) [29], which shares ancestral components with Northern and Eastern African populations. Our goal is to assess how a pangenomic approach enhances the evolutionary and epidemiological characterisation of genetic variation in underrepresented Horn of Africa (HOA) populations, particularly Somalis, by incorporating closely related populations as proxies. Unlike a linear reference genome, which represents a single, fixed sequence, a pangenome (or variation graph) captures the genetic diversity of multiple populations by integrating variations, such as single nucleotide polymorphisms (SNPs), insertions, deletions, and structural changes, into a graph-based structure. This method allows for a more inclusive and representative model of human genetic variation, reducing reference bias and improving read mapping accuracy, especially for populations with diverse or underrepresented ancestries. By leveraging variation graphs, we can systematically evaluate genetic variation across historically interconnected populations, ensuring that newly generated pangenomes more accurately reflect the true ancestral diversity of these groups.

We hypothesise that our implementation of the variation graph approach will significantly enhance variant discovery and genomic inference compared to hg38-based variant calling. To test this, we employ a combination of coalescent and allele frequency-based methods, aiming to demonstrate the practical advantages of this graph-based framework for East African genomics research [29].

## 2. Materials and Methods

Computational analyses for variation graph construction, alignment, and variant calling were performed on the CSIRO High-Performance Computing (HPC) infrastructure with a total capacity of 24,576 CPU cores and ~238 TB memory.

### 2.1. Building the Pangenome Graph

Due to the limited availability of high-quality genomic data for Eastern and Northern Africans, constructing a pangenome graph directly from such individuals was not feasible. Recognising the unique ancestral components of Somalis (and other HOA groups) as distinct from other African populations, we carefully selected a proxy population.

We initially used a panel of 14 Somali individuals from the Human Origins SNP array, integrated into the Allen Ancient DNA Resource (AADR) [30], as a proxy East African target population. The best source populations were selected via *qpAdm* [31] using 593,000 (after lifting over from hg19 to hg38) AADR SNPs [32]. This analysis identified Mozabites as contributing ~47.0% (3sf) ancestry to Somalis as a proxy source population (Table 1). This was the highest contribution for any population for which whole genome data are currently available (Human Genome Diversity Project), making them a suitable proxy population for the graph construction.

Both the reference genome and associated variations were retrieved from the Human Genome Diversity Project (HGDP) [29]. The reference genome was based on Genome Reference Consortium human genome build 38 (GRCh38), and variations originated from 27 Mozabite individuals included in the HGDP. In total, around 11 M variants were used in the construction of the graph (9 M SNPs and 2 M indels). The construction of the variation graph, encompassing the 22 autosomal chromosomes, was facilitated by the variation graph [28] (vg) software (version 1.52).

Guided by the methodology outlined in Martiniano et al. [33], we implemented a stringent Minor Allele Frequency (MAF) filter, excluding variants with MAF below 0.01. This approach effectively removed extremely rare variants, enabling the graph to capture a comprehensive spectrum of variations while avoiding potential biases introduced by unlikely mutations.

### 2.2. Mapping and Variant Calling

Mozabite individuals served as the foundation for constructing the variation graph. To comprehensively evaluate performance, Bedouin individuals were selected as a test population. This decision is based on their close genetic affinity to Mozabites based on shared ancestry (Table 1), as established by various *qpAdm* models involving both the latter and former as source and target populations, respectively. In our study, this particular step was also necessary due to the lack of additional Northern and East African Whole Genome Data. Approval for data access to 225 low-coverage whole genomes from the study of ancestral history by Pagani et al. [2] and from [18] is currently pending.

Variant calling involved multiple methods for robust comparison. A total of 44 Bedouin individuals (from the HGDP dataset) were mapped to the graph using the “vg map” algorithm. Subsequently, “vg call” was employed to identify variants directly from the resulting graph alignment (GAM) files (in the rest of the document, the data from this method are referred to as “GAM”). Additionally, using the “vg surject” command, these GAM files were converted to BAM format, facilitating variant calling with Genome Analysis Toolkit HaplotypeCaller (version 4.2.4.1) [34] (dataset referred to as “GATK” in the next parts of the paper). For those two methods, reads with a mapping quality (MapQ) of 0 were filtered out. This multifaceted approach enabled us to assess each method’s efficiency and potential biases, allowing a comparison of the new results with those from the original HGDP dataset (referred to as “HGDP” in the paper).

To quantify the impact of reference bias on the reliability of variant calls, the Wilcoxon Rank Sum Test was employed, comparing the differences in allele frequency between variant calls generated by the linear reference and the variation graph. This test was also employed for GAM and GATK to account for differences in the results of downstream analyses from various variant callers along with the graph and linear reference.

### 2.3. Effective Population Size Estimates and Simulations

Using the method Relate [35], we estimated effective population sizes of the Negev Bedouin over time, at generation times of 25 and 28 years [35,36]. The parameters for *Ne* (effective population size) estimation included a mutation rate of 1.25 × 10^−8^, the genetic map specifying recombination rates, and generation times, following the protocol by Speidel et al. [35].

To evaluate and compare the reliability of the *Ne* estimates, we conducted coalescent simulations using the maximum likelihood procedure implemented in FastSimCoal (version 28) [37,38]. The number of simulated DNA samples matched the number of haploid sequences in the original Bedouin data (44 individuals). Input parameters for these sequences included a mutation rate of 1.25 × 10^−8^ and a recombination rate of 1.25 × 10^−8^. Bottleneck timings were set to correspond to those found in the empirical data calculated from *Relate*. The maximum likelihood estimates for *Ne* were obtained from 1000 iterations using the Brent algorithm [37,38].

This analysis quantitatively evaluated the degree to which our variation graph improves *Ne* resolution relative to the hg38 linear reference, based on the proximity of simulated versus observed *Ne* estimates.

### 2.4. Metrics of Variant Call Assessment

We assessed the reliability of variant calls obtained in GAM and GATK datasets against the original HGDP variants. Allele frequencies across all Bedouin individuals present in the HGDP database were calculated using *PLINK 2.0* [39]. The two-sided Wilcoxon Rank Sum Test was used to compare the allele frequency distributions between the three variant sets. *p*-values were calculated for all pairwise comparisons.

Additionally, we focused on variants identified as significant hits in a previous GWAS on triglyceride levels in a Kuwait-based Bedouin population [40]. Once again, a Wilcoxon Rank Sum Test was performed to compare the allele frequencies between those datasets. The test was conducted for all significant hits, as well as for those with an allele frequency of less than 0.05 in at least one population within the Genome Aggregation Database (gnomAD) and/or HGDP databases.

## 3. Results

### 3.1. Ancestry Analysis for Variation Graph Construction

To determine the best populations for constructing the variation graph and testing its accuracy on a target population, we used *qpAdm* [31,41]. The results (Table 1) showed that the best-fitting ancestral populations for the Somali target were ancient Kenyans (3300 BP) and Mozabites from the Human Genome Diversity Project (HGDP), with *p* > 0.05. In contrast, modern Kenyans from the 1000 Genomes Project [42] (LWK) had a poor ancestral fit, likely due to the minimal Bantu migration ancestry in the Somali population compared to other East African populations.

Based on these findings, we selected Mozabites from the HGDP as they provide a diverse and representative set of variants for constructing the variation graph. This corroborates the F_ST_ and PCA results in Appendix A, reflective of geographic distributions in Appendix A. The *qpAdm* models also revealed strong evidence of shared ancestry between the Bedouin and Mozabites, making the Bedouin an ideal test population for assessing the pangenomic approach.

### 3.2. Analysis of Variant Caller

To identify similarities and differences between the linear reference and genome graph approaches, as well as the GATK (Genome Analysis Toolkit) and the built-in vg variant callers (using GAM file format), we quantified and compared the detected SNPs and INDELs in terms of quantity and percentage (see Figure 1). We also compared original results from the HGDP publicly available dataset. Although the GAM and GATK datasets were mapped using the same software (vg), they employed different variant callers. Both were generated by filtering out reads with a mapping quality (MapQ) of 0. Due to the different variant calling software, we examined variations based on the SNP quality score (QUAL field in VCF) instead of MapQ. The QUAL score is a phred-scaled measure of confidence in the alternate allele call, where a score of 10 corresponds to a 1 in 10 chance of an incorrect base call (90% accuracy). We analysed how variant counts and percentages differed across QUAL thresholds of 0, 10, 100, 1000, and 10,000.

Using the same filtering, GAM and GATK yielded similar total variant counts (229 M and 231 M, respectively), while HGDP had far fewer at 16 M. This discrepancy stems from HGDP’s more stringent filtering criteria [43], evident when we observe the counts of variants across increasing QUAL thresholds. Consequently, HGDP’s VCFs contained fewer variants but with much higher confidence and with most of their total called variants having high SNP Quality (86.2% > 1000 and 65.4% > 10,000) compared to GAM (9.1% > 1000 and 0.13% > 10,000) and GATK (32% > 1000 and 0.08% > 10,000).

In Figure 1 we can see that the GAM dataset consistently captured a higher percentage of INDELs than GATK and HGDP (except for HGDP at QUAL > 1000), reflecting pangenomes’ ability to better detect INDELs [44]. Apart from this, GAM and GATK exhibited comparable behavior except when considering QUAL > 1000, where 91% of GAM’s variants were filtered out.

### 3.3. Effective Population Size

To evaluate the impact of reference bias on demographic history inferences, we compared the effective population size (*Ne*) estimates from *Relate* [35] for the original Bedouin variant calls in HGDP with those obtained using GAM and GATK (see Figure 2). The latter two were generated using the pangenomic approach. Figure 2 clearly shows that the variant calls in the GAM datasets yield a much lower present-day *Ne* than the HGDP calls, with a continuous decline since the Out-of-Africa migration, reaching a value below 50 individuals at generation times (g) of 25 and 28 years (henceforth labelled as g = 25 and g = 28). A modest increase from the minimum *Ne* levels (after the most recent bottleneck) to present-day levels (~125 individuals) in the last 1500 years is observed only at g = 28, resulting in a present-day *Ne* of approximately 16 individuals for g = 25.

More extreme declines (from an inferred bottleneck 1000 years ago, consistent with the GAM dataset at g = 25 years) are observed using the GATK dataset, with *Ne* dropping below 10 individuals. However, no post-bottleneck population size recovery is observed at either generation time. In contrast, the original HGDP variant calls indicate that the effective population size of the Bedouin increased to a present-day *Ne* of approximately 79,168 (g = 25 years) and 110,485 (g = 28 years) after a bottleneck around 70,000 years ago, approximating the consensus estimated time of the first Out-of-Africa migrations.

Coalescent simulations using FastSimCoal (version 28) indicated that the maximum likelihood estimates of the present-day effective population sizes, assuming generation times of 25 and 28 years, were approximately 17 (simulated 95% CI: 16–19) and 34 (simulated 95% CI: 28–33). These simulated values are for the GAM and GATK datasets, respectively (see Table 2). The simulated *Ne* values represent haploid individuals after 500 likelihood iterations. The *Ne* observations from the GAM dataset yielded values more similar to those detected from simulated data, with the empirical estimate falling within the confidence interval of simulated data at a generation time of 25 years. This contrasts with the variant calls obtained from the hg38 linear reference. While the GATK variant call set performed better than the linear reference, the empirical *Ne* estimate did not fall within the confidence interval of the FastSimCoal simulations.

These results suggest that the GAM approach offers the most reliable variant data for resolving human demography in the Bedouin samples in our study. While this evidence, along with prior knowledge of human evolution, demonstrates the superior performance of the variation graph approach for demographic inference, the absence of data from Pagani et al. [2] limits the accuracy of *Ne* estimation. Nevertheless, our findings caution researchers about the impacts of reference bias on evolutionary and epidemiological analyses in populations with complex diversity, which has been seemingly neglected in present human genetics studies.

To further corroborate the reliability of the *Ne* estimates in the Bedouin and the plausibility of our hypothesis, we examined the differences in allele frequencies between variant calls derived from the variation graph (GAM and GATK) and the original HGDP calls (see Table 3). We observed that both GAM and GATK had significantly increased allele frequencies (Wilcoxon Rank Sum Test *p*-value < 2.2 × 10^−16^) compared to the original HGDP calls, complementing the observed *Ne* patterns. Significant differences were also observed when comparing GAM against GATK but with higher *p*-values than those obtained when comparing each of these call sets against the hg38 calls.

Hence, it is unsurprising that GWAS hits among Kuwaiti Bedouins in a previous triglyceride study [40] (with MAF < 0.05 in at least one HGDP or gnomAD population) have significantly lower frequencies in GAM than the original HGDP variants (*p*-value Wilcoxon = 0.023). This effect was not observed when including all associated variants (Wilcoxon *p*-value = 0.29). However, the distribution shift that occurred has important implications for future GWAS studies in this population.

## 4. Discussion

In this study, we compared the pangenomic approach using variation graphs to the traditional linear reference approach to assess their effectiveness in variant detection. While both GAM and GATK datasets exhibited comparable overall variant detection in terms of counts and types, GAM identified approximately 2 million more insertions and deletions (INDELs) than GATK, consistently showing a higher percentage of INDELs across most quality thresholds. This enhanced INDEL sensitivity aligns with the well-established advantage of variation graphs in detecting structural variants [44]. However, as quality thresholds increased beyond 100 and 1000, GATK’s INDEL counts surpassed those of GAM. Ultimately, the HGDP dataset demonstrated the highest confidence variant calls compared to our simple mapping quality (MapQ) > 0 filtering for GAM and GATK, reflecting its stringent criteria and reputation as a high-quality resource relied upon by researchers.

Our findings highlight how the variation graph approach enhances the study of human evolutionary history in underrepresented populations by integrating well-established population genetics methods. The use of *qpAdm* for ancestral modelling played a crucial role in identifying the optimal source and target populations, allowing us to efficiently fit genetic relationships using a likelihood ratio test approach [31,41]. Selecting the HGDP Mozabite population for building the variation graph, based on *qpAdm* results (Table 1), was useful for our purposes in terms of performance gains over the linear reference. A streamlined process to implement custom-built graphs in genomic medicine as performed here is essential. This re-emphasises the intuitive discoveries by Tetikol et al. [27], concluding that variation graphs with greater population specificity are far superior to more generic alternatives or proxies such as the Human Pangenome Reference Consortium (HPRC) [45] and the draft Arab Pangenome Reference (APR) [46]. The apparent disparity with the latter example is presumably magnified by the unique way in which evolutionary forces influenced genetic diversity in North Africa and the HOA [9,12,18,47]. Our results emphasise the need for custom-built, population-specific graphs to enhance genomic medicine and evolutionary research in diverse human populations.

The >10,000-fold discrepancy in modern-day effective population size (*Ne*) estimates between the GAM and GATK variant datasets and the original HGDP dataset, along with the significantly deviated allele frequency distributions in the latter (*p*-value < 2.2 × 10^−16^), may be due to excessive reference bias in the linear reference genome. Reference bias can introduce false-positive rare variants, which, in turn, reduce coalescence rates and inflate population size estimates [48,49,50]. This effect may be particularly pronounced in the Bedouin population, given its complex population structure and admixture patterns [51]. While additional data and sequence read simulations would be necessary to confirm this hypothesis, the alignment of our observed *Ne* values with simulated inferences from FastSimCoal, along with the well-documented effects of reference bias, supports this explanation for our *Ne* estimates and allele frequency patterns.

In addition to the coalescent simulations producing present-day *Ne* estimates that align with the 95% confidence interval for GAM variant calls, our findings are strongly supported by the existing literature on the genomic and anthropological history of the Bedouin population. Historically, Bedouin groups have been considered a genetic isolate, with significant barriers to intermarriage and documented occurrences of consanguineous marriages [52,53,54]. Population genetic studies have shown that Bedouins, particularly those from Kuwait, display distinct principal component analysis (PCA) and F_ST_ clustering patterns compared to other non-African populations. They also exhibit greater interpopulation distances relative to their Persian Gulf neighbors, likely due to high levels of genetic drift [52]. Additionally, evidence points to ancient bottleneck events, an early divergence from African populations following the initial Out-of-Africa migrations, and lower levels of Neanderthal admixture compared to Europeans and Asians [6].

In the context of prior research, our findings are particularly significant given the scarcity of studies using pangenomic methods to analyse Bedouin, North African, and Middle Eastern genetic data. To our knowledge, this is the first study to demonstrate the potential of such an approach in mitigating reference bias and its direct impact on demographic history analyses in a Middle Eastern population. This work also has broader implications for understanding East African and Horn of Africa (HOA) human genetics. However, we acknowledge the substantial contributions of large consortia, such as TopMed, toward similar objectives.

Recently developed imputation reference panels [55,56] include individuals from Northeast Africa, meaning that whole genome sequencing data, currently restricted without the necessary permissions, could be used for variation graph construction. However, these datasets have several limitations. For example, 23 and Me’s local ancestry inference algorithm, which uses support vector machines and a hidden Markov model, estimates that Northern East African (encompasses the HOA) individuals contribute only 1.1% (on average) of the ancestry in the aggregated African American (AFAM) dataset from TopMed [55]. This likely reflects sample size imbalances, which reduce both the number of individuals available and the overall power for variation graph generation and testing. By contrast, across the same dataset, average Atlantic West African ancestry is estimated at 66%, with over 2000 individuals having more than 50% of their genetic composition coming from this population group. Therefore, most AFAM panel individuals largely appear to be descendants of source populations that contributed minimally to the genetic history of present-day HOA Ethiopians and Somalis [55].

Furthermore, as expected from ancestry deconvolution analyses, the majority of the AFAM panel clusters with West Africans from the 1000 Genomes Project (as shown by UMAP analysis), making their genetic diversity incompatible with population genetics and epidemiological studies focused on HOA populations. A similar issue arises with the imputation panels analysed by Sengupta et al. [56], where the East African individuals included are primarily from modern-day Kenya, again, genetically distinct from the HOA.

Many researchers have emphasised that this discrepancy cannot be explained solely by differences in Eurasian gene flow [3,12,55], as suggested by Sengupta et al. (2023) [56]. Instead, another major factor is that the HOA was largely unaffected by allele frequency shifts associated with the Bantu migration. Without a more refined sampling strategy that captures the fine-scale genetic substructure shaped by both within- and transcontinental migrations across Africa, current panels from TopMed will likely yield unreliable inferences for populations in this region.

The small number of individuals identified as part of a unique Northeast African genetic cluster are inferred to have approximately 33% Somali-related ancestry, with the remainder predominantly assigned to Ethiopian and Eritrean lineages. As previously noted, HOA populations exhibit extensive within-region genetic differentiation, often exceeding that of some Maghreb populations, which include some of the most genetically isolated groups since the Paleolithic [18,19]. This distinction is further highlighted by ADMIXTURE results from Hodgson et al. [12], which show that Ethiopian-based Somali populations possess a significantly higher proportion of a non-Indigenous (Ethio-Somali) genetic component compared to Ethiopian groups such as the Ari and Amhara [12]. Unlike the Maghreb and the Arabian Peninsula, this component represents a unique genome-wide ancestry profile within the HOA. Nonetheless, with current efforts underway to mitigate this sampling gap, it is hoped that resources such as the TopMed imputation server will provide more knowledge into HOA genetics.

While our study provides promising insights into the effectiveness of variation graphs in mitigating reference bias and improving demographic history analyses, certain limitations remain. Specifically, the lack of access to East African genetic data from Pagani et al. and Lucas-Sanchez et al. (2024) [2,18] prevented us from conducting comprehensive simulations that account for historical migration and admixture between Africa and Eurasia. Such simulations would enable a more precise estimation of ancestry-specific *Ne* and other key demographic parameters [57]. However, our primary objective was to assess the impact of reference bias on the genetic history of populations with shared ancestry among Eastern and Northern Africans, rather than to develop high-resolution demographic models of the Bedouin at a fine-scale genomic level. Nevertheless, once the requested data become available, comparative analyses of our variation graph with the draft APR [46] and other existing graph-based reference models will become more feasible, further enhancing the downstream applications of this work.

Our analyses also have important implications for medical genetics. The lower frequency of Bedouin variant calls in the GAM dataset compared to the original HGDP dataset for genome-wide association study (GWAS) hits from a previous study [40] raises concerns about potential false-positive associations. Similar investigations into GWAS allele frequencies are crucial for advancing genomic medicine, particularly in populations with complex ancestries, such as those represented in the Qatar Biobank, and should be prioritised in future research [58,59]. The diversity of these individuals, along with the ancestral affinity some share with the Bedouin, suggests they also possess a complex population structure and history shaped by shifting genetic connections with Northern and Eastern Africans over time [20,58,59,60].

In summary, our findings highlight the potential of the variation graph approach not only in mitigating reference bias and improving demographic history inferences but also in enhancing medical genetics studies for underrepresented populations with complex ancestries.

## 5. Conclusions

Our study highlights the value of variation graphs in uncovering population dynamics and genetic diversity in underrepresented groups, such as the Bedouin. By incorporating variation graphs into population genetics, we challenge existing perspectives on human history and genome-wide association studies (GWAS) in Middle Eastern populations. Pangenomic approaches mitigate reference bias, offering a more accurate representation of the genetic architecture of East Africans and other diverse populations. As more genomic data from East Africa becomes available, variation graphs hold significant potential for reducing disparities in evolutionary and translational medicine by enabling a more precise characterisation of populations with complex ancestries.

## Figures and Tables

**Figure 1 biomolecules-15-00582-f001:**
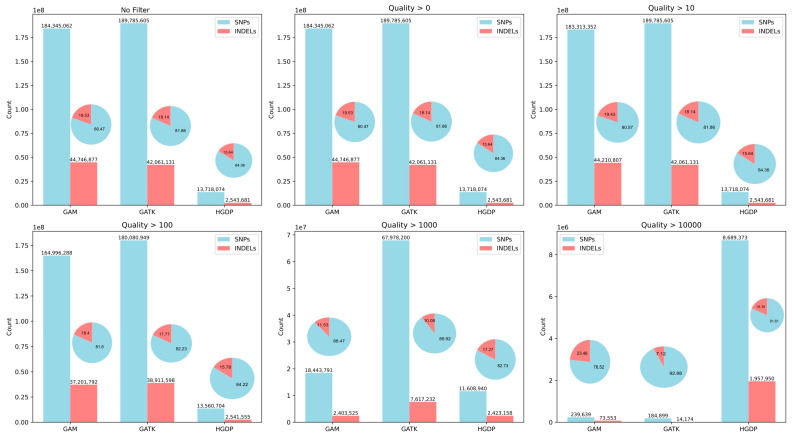
Variant types and counts across SNP quality thresholds for each dataset. The bar plots show the number of SNP and INDEL variants detected using the GAM, GATK, and HGDP datasets. The pie charts represent the percentage of each variant type (SNPs or INDELs) detected for each method under a given quality threshold. The subplots display variant counts after applying different SNP quality score (QUAL) filters, ranging from no filtering (**top left**) to stringent filtering (QUAL > 10,000, **bottom right**).

**Figure 2 biomolecules-15-00582-f002:**
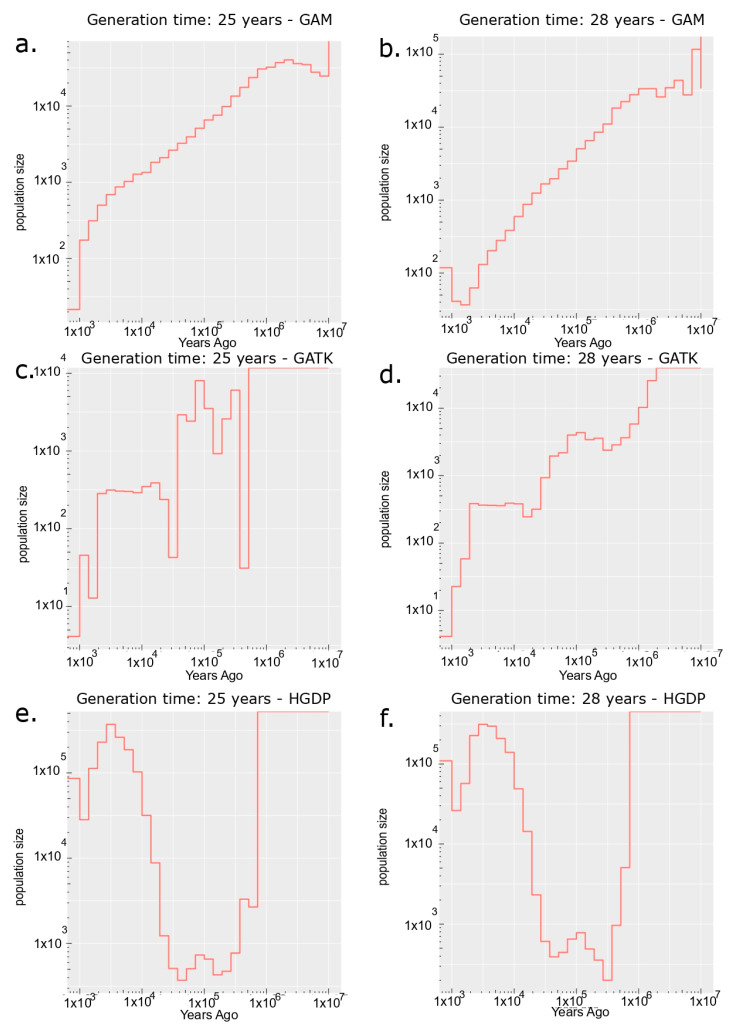
Bedouin population size estimates across all datasets. Effective population size over time for the Bedouin population estimated using the coalescent-based *Relate* method. Formulas for converting coalescence rates to effective population sizes can be found and implemented from the original *Relate* manuscript and documentation [35]. Subplot titles indicate the dataset used and generation time assumptions. Information for each subplot is as follows: Effective population size estimates are given for generation times of 25 and 28 years respectively for the GAM (**a**,**b**), GATK (**c**,**d**) and HGDP (**e**,**f**) variant datasets.

**Table 1 biomolecules-15-00582-t001:** *qpAdm* ancestry model fits for source and target populations.

Target	Source Populations	*p*-Value
Somali	Kenya Kansyore 3300 BP (53.0%), Mozabite (47.0%)	0.524
Somali	Kenya_IA_Pastoral (27.0%), Ethiopian_BetaIsrael (Ethiopian Jews) (73.0%)	0.189
Somali	LWK (86.7%), Mozabite (5.2%) Kenya IA (8.1%)	3.08 × 10^−46^
Mozabite	MSL (31.8%), Palestinian (68.2%)	0.347
Mozabite	MSL (19.0%), Bedouin (81.0%)	0.453
Bedouin	Palestinian (69.0%), Mozabite (31.0%)	0.377
Bedouin	Palestinian (62.0%), Mozabite (38.0%)	0.389

Results from *qpAdm* modelling to infer the best source population combinations that explain the ancestral composition of target populations. Well-supported models have *p*-values > 0.05. The target populations are Somalis from the Human Origins Array and Bedouins from the Human Genome Diversity Project (HGDP). The Bedouins are used as a proxy for testing variant call reliability in the absence of available whole genomes from East African populations. Their inclusion is justified by the *qpAdm* ancestry composition output, which shows shared ancestry with the source populations used for graph construction.

**Table 2 biomolecules-15-00582-t002:** Estimated effective population sizes for all datasets from empirical and simulated data using *Relate* and *FastSimCoal*.

Dataset	Generation Time	*Ne* Estimation: Empirical Data (*Relate*)	*Ne* Simulated Estimation: FastSimCoal v28 (95% CI)
GAM	25	16	17 (95% CI: 15–19)
GAM	28	125	34 (95% CI: 28–35)
HGDP	25	79,168	3295 (95% CI: 324–3468)
HGDP	28	110,485	9639 (95% CI: 5164–78,179)
GATK	25	4	13 (95% CI: 12–14)
GATK	28	3	14 (95% CI: 11–15)

Present-day effective population size (*Ne*) estimates (rounded to the nearest integer) from Relate [19] for the Bedouin population, using variants from GAM and GATK (both graph-based approaches), and the original HGDP dataset. It also includes simulated *Ne* estimates with 95% confidence intervals from *FastSimCoal* v28, calculated at generation times of 25 and 28 years.

**Table 3 biomolecules-15-00582-t003:** Wilcoxon tests for allele frequency differences.

Variant Allele Frequency Comparisons	*p*-Value (Direction of Frequency Distribution Shift)
GAM vs. GATK	1.76 × 10^−3^ (GAM > GATK)
GAM vs. HGDP	<2.2 × 10^−16^ (GAM > HGDP)
GATK vs. HGDP	<2.2 × 10^−16^ (GATK > HGDP)

Results of Wilcoxon Rank Sum Tests comparing allele frequency distributions between variant calls from the *vg* graph-based approach (GAM and GATK datasets) versus the original HGDP calls. For each comparison, the *p*-value is provided, with the direction of the relative allele frequency shift indicated in parentheses.

## Data Availability

The genomic datasets used in this study were obtained from the Human Genome Diversity Project: HGDP|IGSR data collection (internationalgenome.org). The variation graph files generated during this study are available in the repository hosted at: Index of /users/1269082/graph (csiro.au) (accessed on 9 April 2025).

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
