# Peer review of "A Pangenomic Approach to Improve Population Genetics Analysis and Reference Bias in Underrepresented Middle Eastern and Horn of Africa Populations"

_biomolecules, 2025, doi:10.3390/biom15040582_

Round 1
Reviewer 1 Report
Comments and Suggestions for Authors
This manuscript explores one of the most relevant topics of interest in whole genome sequencing in populations underrepresented in database repositories (as HGDP). One of the questions that arise in studies in genomic medicine in non-European populations is which is the best reference for sequencing the human genome in these underrepresented populations and whether it is possible to sequence low-frequency genetic variables, especially in populations of African origin where it is known that the populations with the greatest diversity in the world exist and where the creation of the pangenomics project is justified to have reference genomes that serve as a better reference than the Genome Reference Consortium build 38 (GRChg38) of the human genome compared to the Equations for the effective size (Ne) method for the sequencing of underrepresented populations.
This manuscript suggests clarifying and expanding additional information that should be included to be considered for publication:
- The first comment and suggestion on this manuscript is that the first section is the introduction (which the authors title "background") section instead of doing the method section first. I suggest that the authors re-order the sections in a coherent order of ideas so that readers can properly follow the sequence of this study.
- The authors introduce human groups in Africa such as the Somali population, the Mozabites and the Bedouins. a) It is suggested to include a graphical description of these groups using a map (of Africa) of where these groups are located, as well as being able to run a principal component analysis (PCA) comparing these groups with other African groups to put the use of these populations into context. b) It is also suggested to include a table of the Fst statistics and a PCA figure to provide a quantitative description (genomic distance) of these African groups. (in the supplementary information section)
- In the last paragraph the authors write: “This study aims to quantify the impact of the pangenomic approach on the evolutionary and epidemiological characterization of genetic variation in underrepresented East African populations by incorporating populations with shared ancestral components as proxies.” in this objective, authors are encouraged to discuss their opinion (and limitations) on variant imputation methodologies using tools such as the TopMed Imputation Server tool as reference information to impute low-frequency variants in underrepresented populations of African origin (Bedouin or individuals with Somali ancestry) and whether this method is useful (or not) in genomic medicine studies compared to the pan-genomic approach.
- Tables 2, 3 and Figure 1 do not appear to be cited in the manuscript. Authors are advised to ensure that citations for them are included in the appropriate text.
- This manuscript is 66% similar to the preprint version, https://assets-eu.researchsquare.com/files/rs-4651266/v1_covered_559bdb84-a827-42b0-83b1-8b424b0a301b.pdf?c=1722514594 it is suggested to withdraw the preprint version if this work is accepted.
Authors are encouraged to generate a new version of the manuscript taking into consideration the suggestions made by the reviewers and to review the quality and format of the tables and figures.
Author Response
We sincerely appreciate the time and effort the reviewers have dedicated to evaluating our manuscript. Their insightful comments and suggestions have been carefully considered, and we believe they have helped strengthen our work. Below, we provide detailed responses to each point raised:
- The first comment and suggestion on this manuscript is that the first section is the introduction (which the authors title "background") section instead of doing the method section first. I suggest that the authors re-order the sections in a coherent order of ideas so that readers can properly follow the sequence of this study.
We sincerely thank the reviewers for their valuable comments. In response, we have reorganized the manuscript into the following structure: Background,Materials and Methods, Results, Discussion, and Conclusion
2. The authors introduce human groups in Africa such as the Somali population, the Mozabites and the Bedouins. a) It is suggested to include a graphical description of these groups using a map (of Africa) of where these groups are located, as well as being able to run a principal component analysis (PCA) comparing these groups with other African groups to put the use of these populations into context. b) It is also suggested to include a table of the Fst statistics and a PCA figure to provide a quantitative description (genomic distance) of these African groups. (in the supplementary information section)
We thank the reviewer for this comment and realise the effort in understanding the requirements for including the multiple different populations included in this manuscript. To provide clarity, we have conducted additional analysis and representations as a compliment to the analysis already provided in the results. PCA and FST for all populations are now included in the supplementary materials (see Figure S1 and Table S1, respectively), along with explanatory text. Additionally, as requested, we have added a map showing the locations of all populations (Figure S2 in the supplementary material).
Those new results are referred in the text in the Results part of the manuscript (see Ancestry Analysis for Variation Graph Construction)
3. In the last paragraph the authors write: “This study aims to quantify the impact of the pangenomic approach on the evolutionary and epidemiological characterization of genetic variation in underrepresented East African populations by incorporating populations with shared ancestral components as proxies.” in this objective, authors are encouraged to discuss their opinion (and limitations) on variant imputation methodologies using tools such as the TopMed Imputation Server tool as reference information to impute low-frequency variants in underrepresented populations of African origin (Bedouin or individuals with Somali ancestry) and whether this method is useful (or not) in genomic medicine studies compared to the pan-genomic approach.
We appreciate the reviewer’s insightful comment regarding the use of TopMed and variant imputation methodologies. While the TopMed initiative represents a significant contribution to genomic research, we have discussed the limitations of its applicability to populations with complex substructure, such as those of African origin, including Bedouins and individuals with Somali ancestry. As highlighted in our manuscript, genetic variation across African groups is shaped by extensive substructure, resulting from shifts in ancestry and allele frequency dynamics over time from multiple factors, mainly through differing genetic drift and admixture occurrences. .
In response to the reviewer’s suggestion, we have elaborated on TopMed approach and the possible issues in the discussion section, citing key studies, including O’Connell et al. 2021, Sengupta et al. 2023, and other relevant works on the Horn of Africa (HOA) populations (e.g., Pagani et al. 2015, Hodgson et al. 2014). These references help underline the challenges in applying imputation methodologies like those used by TopMed to populations with such complex genetic histories.
The discussion section should provide clarity on the potential limitations of using TopMed panels for underrepresented populations and the advantages of the pangenomic approach in these contexts.
4. Tables 2, 3 and Figure 1 do not appear to be cited in the manuscript. Authors are advised to ensure that citations for them are included in the appropriate text.
We thank the reviewers for identifying this issue. As requested, we have added references to Figure 1 where necessary in the manuscript (in the Analysis of Variant Callers section). Similarly, we have included references to Table 2 and Table 3 in the Effective Population Size section
5. This manuscript is 66% similar to the preprint version, https://assets-eu.researchsquare.com/files/rs-4651266/v1_covered_559bdb84-a827-42b0-83b1-8b424b0a301b.pdf?c=1722514594 it is suggested to withdraw the preprint version if this work is accepted.
Yes, this is our plan. However, we will wait for the manuscript to be formally accepted before withdrawing the preprint. In addition, we have substantially revised the background and discussion to differentiate this version from the preprint, incorporating additional information on population history. We have also expanded the results section to address the reviewers' previous comments.
Reviewer 2 Report
Comments and Suggestions for Authors
The manuscript is difficult for reading and reviewing as it was submitted with tiny letters due to the comment field on the right included. So, this review includes comments on the issues I can see and understand in the text and figures/tables.
The manuscript has a very unusual structure starting with Methods and followed by Background. However, Background did not provide good explanation on the history and genetics of the populations studied and on approaches how to construct pangenomic sequences, making it difficult for readers to understand the previous studies on these topics. I checked a pair of articles in Biomolecules and they have a standard structure with Introduction first followed by Materials and Methods.
There is huge confusion with the usage of different ethnic groups. Negev Bedouins used in the title were mentioned only one time in the manuscript text on Line 92 – 93. In other places in the manuscript, readers meet different populations including Somalis, Mozabites, Bedouins, Kuwait-based Bedouins, Bedouin from the Human Genome Diversity Project, ancient Kenyans, modern Kenyans, some other groups in Table 1. It is necessary to describe all these groups and how they are related through the human evolution and migrations in Africa and the Near East and finally focus the manuscript on Negev Bedouins.
The manuscript looks like a technical report. The manuscript compared the pangenomic variation graph approach and the traditional linear graph approach to count the number of SNPs and indels in a few datasets. It also evaluated the changes in effective population sizes through some period for two different generation times. I cannot review that topic because Figure 2 is unreadable.
The title looks misleading because the focus of the manuscript on population genetics of Negev Bedouins and their genomic medicine is uncertain.
Author Response
We sincerely appreciate the time and effort the reviewers have dedicated to evaluating our manuscript. Their insightful comments and suggestions have been carefully considered, and we believe they have helped strengthen our work. Below, we provide detailed responses to each point raised.
1- The manuscript is difficult for reading and reviewing as it was submitted with tiny letters due to the comment field on the right included. So, this review includes comments on the issues I can see and understand in the text and figures/tables.
We regret the formatting issue that made the manuscript difficult to read. However, this was due to the submission process and not within our control.
2- The manuscript has a very unusual structure starting with Methods and followed by Background. However, Background did not provide good explanation on the history and genetics of the populations studied and on approaches how to construct pangenomic sequences, making it difficult for readers to understand the previous studies on these topics.
We appreciate this important suggestion. To address this, we have fully rewritten the introduction / background section, integrating a more detailed description on the genetic and archaeological history of the studied populations. This revision provides a clearer contextual foundation, as requested. While we acknowledge the increased length, we believe the additional details enhance the manuscript by offering a comprehensive history of these populations alongside an introduction to pangenomics.
Furthermore, we have added an explicit explanation of how pangenomes work in the background section:
"Unlike a linear reference genome, which represents a single, fixed sequence, a pangenome (or variation graph) captures the genetic diversity of multiple populations by integrating variations, such as single nucleotide polymorphisms (SNPs), insertions, deletions, and structural changes, into a graph-based structure. This approach provides a more inclusive and representative model of human genetic variation, reducing reference bias and improving read mapping accuracy, particularly for populations with diverse or underrepresented ancestries. By leveraging variation graphs, we can systematically evaluate genetic variation across historically interconnected populations, ensuring that newly generated pangenomes more accurately reflect the true ancestral diversity of these groups."
For further details on pangenomics and the specific software we used (VG), we refer to the cited work by Garrison et al. We hope these revisions improve clarity regarding previous studies on these topics and strengthen the rationale behind our objectives and conclusions.
3- I checked a pair of articles in Biomolecules and they have a standard structure with Introduction first followed by Materials and Methods.
We sincerely thank the reviewers for their valuable comments. In response, we have reorganized the manuscript into the following structure: Background, Materials and Methods, Results, Discussion, and Conclusion
4- There is huge confusion with the usage of different ethnic groups. Negev Bedouins used in the title were mentioned only one time in the manuscript text on Line 92 – 93. In other places in the manuscript, readers meet different populations including Somalis, Mozabites, Bedouins, Kuwait-based Bedouins, Bedouin from the Human Genome Diversity Project, ancient Kenyans, modern Kenyans, some other groups in Table 1. It is necessary to describe all these groups and how they are related through the human evolution and migrations in Africa and the Near East and finally focus the manuscript on Negev Bedouins.
We appreciate this important feedback. To address this, we have fully rewritten the introduction to provide a more detailed historical background on the studied populations. This revised section explains the evolutionary relationships between these groups over time, drawing on published research to clarify their relevance to our study. Additionally, we have expanded the discussion to further elaborate on these connections, incorporating revisions based on previous reviewer comments.
We would also like to clarify that the Negev Bedouins are the same as the Bedouins from the HGDP dataset, and they are referenced 33 times throughout the manuscript (excluding figures and captions). To minimize confusion, we have carefully revised the manuscript to ensure consistent population labels and clearer distinctions where necessary.
5- The manuscript looks like a technical report. The manuscript compared the pangenomic variation graph approach and the traditional linear graph approach to count the number of SNPs and indels in a few datasets. It also evaluated the changes in effective population sizes through some period for two different generation times. I cannot review that topic because Figure 2 is unreadable.
We thank the reviewers for highlighting the readability issue in Figure 2. We have increased the size of the legend and axis labels and ensured consistent nomenclature throughout.
6- The title looks misleading because the focus of the manuscript on population genetics of Negev Bedouins and their genomic medicine is uncertain.
We sincerely thank the reviewers for their feedback. We have revised the title to better reflect the study’s focus on reference bias rather than medical implications. The new title is:
A Pangenomic Approach to Improve Population Genetics Inference and Reference Bias in Underrepresented Middle Eastern and Horn of Africa Populations
This change ensures that the manuscript accurately conveys its primary emphasis on reference bias and population genetics. We appreciate the reviewers' suggestion, which has helped improve the clarity of our work.
Round 2
Reviewer 1 Report
Comments and Suggestions for Authors
Thank you for properly addressing the suggestions made in the first review. I just want to make one last minor suggestion regarding the first paragraph on page 2 (lines 58-61). You have a statement in parentheses: "(after controlling for recent introgression within the past 1–2 thousand years)" Regarding this statement, I am not sure if you are quoting information from reference [2] or if this data is a calculation made by you. (the authors of this manuscript).
Author Response
We thank the Reviewer for highlighting this point. To clarify, this part of the sentence referred to an analysis conducted by the first author of the manuscript cited as reference 2, and is not part of the results presented in this study. We agree that the original sentence was unclear. To avoid confusion, we have deleted this portion, which we believe improves the clarity of the text. If the Reviewer is still interested in this analysis, it is thoroughly described in the Methods and Results sections of reference 2.